# Nutritional and Supplemental Interventions for Prevention and Treatment of Oral Mucositis in Pediatric Oncology

**DOI:** 10.3390/nu17223521

**Published:** 2025-11-11

**Authors:** Razvan Mihai Horhat, Alexandru Alexandru, Cristiana-Smaranda Ivan, Norberth-Istvan Varga, Madalina-Ianca Suba, Elena Ciurariu, Monica Susan, Razvan Susan, Adrian Cote

**Affiliations:** 1Department of Restorative Dentistry, Faculty of Dentistry, “Victor Babes” University of Medicine and Pharmacy, Eftimie Murgu Square No. 2, 300041 Timisoara, Romania; horhat.razvan@umft.ro; 2Doctoral School, “Victor Babes” University of Medicine and Pharmacy, Eftimie Murgu Square No. 2, 300041 Timisoara, Romania; smaranda.ivan@umft.ro (C.-S.I.); norberth.varga@umft.ro (N.-I.V.);; 3Multidisciplinary Research Center on Antimicrobial Resistance (MULTI-REZ), Microbiology Department, “Victor Babes” University of Medicine and Pharmacy, 300041 Timisoara, Romania; 4Methodological and Infectious Diseases Research Center, Department of Infectious Diseases, “Victor Babes” University of Medicine and Pharmacy, 300041 Timisoara, Romania; 5Discipline of Physiology, Department of Functional Sciences III, “Victor Babes” University of Medicine and Pharmacy, Eftimie Murgu Square No. 2, 300041 Timisoara, Romania; 6Discipline of Immunology and Allergology, Biology, Department of Functional Sciences III, “Victor Babes” University of Medicine and Pharmacy, Eftimie Murgu Sq. No. 2, 300041 Timisoara, Romania; 7Department of Internal Medicine I, Centre for Preventive Medicine, “Victor Babes” University of Medicine and Pharmacy, Eftimie Murgu Square No. 2, 300041 Timisoara, Romania; 8Department of Family Medicine, Centre for Preventive Medicine, “Victor Babes” University of Medicine and Pharmacy, Eftimie Murgu Square No. 2, 300041 Timisoara, Romania; 9Department of Surgical Disciplines, Faculty of Medicine and Pharmacy, University of Oradea, 410073 Oradea, Romania

**Keywords:** oral mucositis, mucositis, pediatric cancer, pediatric oncology, nutrition, honey, medical grade honey, vitamin E, Aloe vera, olive oil

## Abstract

*Background*: Oral mucositis (OM) is a frequent complication of anticancer therapy which arises from cytotoxic injury, having significant clinical implications. Nutritional and supplement-based interventions have been proposed as adjunctive strategies to improve outcomes. *Objectives*: This systematic review aimed to identify and synthesize evidence from randomized controlled trials (RCTs) evaluating nutritional or natural supplement interventions for prevention or management of OM in pediatric oncology. *Methods*: We conducted a systematic search (17 August 2025) of Scopus, PubMed/MEDLINE, and Google Scholar (1 January 2000–1 June 2025) following PRISMA guidelines and registered in PROSPERO (CRD420251134454). The review included randomized controlled trials in pediatric cancer patients (≤18 years; up to 25 years for follow-up) receiving chemo-/radiotherapy, assessing nutritional, dietary, or natural product interventions for oral mucositis prevention or treatment. Non-randomized, adult, non-English, non-peer-reviewed, or inaccessible studies were excluded. Outcomes included incidence, severity, duration of OM, and mucositis-associated pain. Risk of bias was assessed using the NIH Study Quality Assessment Tools and the Cochrane RoB 2 tool. Results were qualitatively summarized. *Results*: Of 5870 records identified, 20 RCTs met inclusion criteria resulting in 1430 total included patients. Interventions tested included systemic supplements (e.g., glutamine, zinc, and bovine colostrum), topically applied agents (e.g., honey, vitamin E, Aloe vera, and olive oil), and nutrient-containing rinses (e.g., chamomile, Caphosol, and Traumeel S). Honey-based interventions showed promising outcomes. *Discussion:* Study designs and sample sizes varied considerably, and outcome measures were heterogeneous. Challenges with blinding, variable compliance, and inconsistent reporting reduce confidence and precision in the findings. *Conclusions*: Evidence from pediatric RCTs remains limited but highlights nutritional and natural products as promising supportive care options for OM. Findings suggest potential for practical, low-cost adjuncts to established oral care protocols, warranting further high-quality multicenter trials.

## 1. Introduction

Oral mucositis (OM) is a frequent and debilitating adverse effect of cancer therapy in children and adolescents, particularly in the context of intensive chemotherapy, radiotherapy, and hematopoietic stem cell transplantation [1,2,3]. Its pathophysiology involves direct cytotoxic injury to rapidly dividing mucosal epithelial cells, amplified by inflammatory and microbial processes. High-dose methotrexate, cytarabine, melphalan, busulfan, and platinum-based agents are among the most common contributors in pediatric regimens [4,5,6]. Clinically, OM manifests as erythema, ulceration, and pain, often leading to impaired oral intake, increased risk of infection, prolonged hospitalization, and treatment interruptions [7].

Oral mucositis arises through a complex succession of events and not direct mucosal injury alone. The generally accepted five-phase model describes (i) initiation through DNA damage and excessive reactive oxygen species (ROS) generation, (ii) upregulation and message generation via transcriptional pathways, (iii) signal amplification through inflammatory mediators, (iv) ulceration with microbial colonization, and (v) healing.

The main causes of OM are the increased presence of reactive oxygen species (ROS) and direct DNA damage, leading to apoptosis of basal epithelial cells [8]. The molecular pathways mediating this process are intricate, with the transcription factor NF-κB being the “gatekeeper” molecule, regulating multiple downstream pathways and over 200 genes including pro-inflammatory cytokines, as well as stress response mediators and notably adhesion molecules and matrix-degrading enzymes [8,9,10,11]. Concomitantly, fibroblast activation and metalloproteinase release degrade extracellular matrix components, weakening mucosal integrity [12]. The subsequent ulcerative stage is clinically evident, with painful breaches of the mucosa that allow microbial invasion, leading to additional inflammation and potential systemic infections [13]. Finally, the healing phase occurs once anticancer treatment subsides, with mesenchymal signals and extracellular matrix stimuli driving tissue re-epithelialization [8].

With that in mind, aiming at just one mechanism to prevent or treat mucositis is unlikely to be sufficient. This is the reason behind the search for practical, easy to implement, potential solutions. Some solutions may be based on known biochemical pathways while others may be based simply on clinical evidence, the key objective being to improve the quality of life of patients.

Despite advances in supportive oncology care, the management of OM remains a significant challenge. Pharmacological interventions have shown limited efficacy, and no universally accepted standard preventive therapy exists in pediatrics [6,14]. Nutritional and supplement-based interventions—such as honey, omega-3 fatty acids, glutamine, probiotics, Aloe vera, and other plant-derived bioactive substances—have been investigated for their potential to modulate inflammation, promote mucosal healing, and reduce symptom burden. While several randomized and observational studies suggest benefits, evidence remains fragmented, heterogeneous in methodology, and largely derived from small cohorts or adult populations [15,16].

The need for separate analysis of the pediatric population from the adult population can be explained through several notable differences:Children’s oral epithelium is thinner and often less keratinized, particularly in the buccal mucosa and tongue, compared to adults’ [17,18].Children’s immune systems are still developing (different T-cell subsets and cytokine profiles), with differences in neutrophil function [19,20].Pediatric oral mucosa cell turnover varies, with a slow neonatal turnover, a high turnover rate in childhood, and a slow turnover rate in adulthood. The regenerative capacity also differs compared to adults [21,22,23].Children’s growth, higher per kg nutrient needs (vitamin intake, amino acid profile, and caloric need), and feeding/taste aversions during therapy make OM a bigger driver of undernutrition and downstream outcomes than in adults [24,25,26,27,28].Children more often receive leukemia/lymphoma regimens and HSCT, while adults dominate head-and-neck RT/CRT—so the dominant mucositis causal agents differ by age. This can also be seen through differences in prevalence rates between the two populations regarding type of therapy involved [29,30,31,32].The core five-phase injury model is shared, but modifiers differ: developing immunity, mucosal turnover, saliva quantity/quality, and microbiome composition in children vs. adults differ and can influence the course of OM (supported by foundational biology) [8,33,34].

Regarding nutritional interventions against oral mucositis, they can be grouped into three categories: systemically administered supplements intended to modulate host metabolism, immunity, and inflammatory responses from within; topical applications that act directly at the site of mucosal injury intended to deliver prolonged local antioxidant and anti-inflammatory effects and promote wound-healing; and nutrient-containing mouth rinses designed to deliver antibacterial substrates while also mechanically removing debris and potential harmful bacteria, with general coverage.

The objective of this review is to systematically identify and synthesize available evidence on nutritional and supplement-based interventions for the prevention or treatment of oral mucositis in children and adolescents (≤18 years, including studies that covered young adult patients who initiated treatment in childhood and were included in pediatric cohorts up to 25 years of age due to follow-up period) with cancer or undergoing hematopoietic stem cell transplantation. Specifically, the scope will include both dietary and topical nutritional supplements used for the prevention or reduction of oral mucositis in pediatric oncology.

To define such dietary supplements, we refer to products ingested orally that provide nutrients or bioactive compounds with a potential protective effect on the oral mucosa, such as vitamins, minerals, amino acids (e.g., glutamine), probiotics, colostrum, and natural products like honey when taken orally. In addition, topical nutritional supplements will also be considered when their active components are nutritional or natural in nature. These include agents applied locally in the oral cavity, such as honey rinses, Aloe vera gel, chamomile mouthwash, propolis solutions, or olive oil rinses. Although applied topically, such interventions are still derived from nutritional or natural sources and are often classified as supplements or nutraceuticals. By contrast, non-nutritional topical drugs (e.g., chlorhexidine, benzydamine, and palifermin rinses) and device-based or purely pharmacological interventions are beyond the scope of this review and will not be included.

To summarize, this systematic review aimed to identify and synthesize evidence from randomized con-trolled trials (RCTs) evaluating nutritional or natural supplement interventions for prevention or management of OM in pediatric oncology.

## 2. Materials and Methods

### 2.1. Guideline and PICO

This systematic review followed the guidelines of the Preferred Reporting Items for Systematic Reviews and Meta-Analyses (PRISMA) 2020 statement [35]. The PRISMA checklist with the requested information is available in the Appendix A. This review was prospectively registered in the PROSPERO international systematic review registry (PROSPERO CRD420251134454).

To ensure clarity and methodological rigor, the review question was developed using the PICO framework (Population, Intervention, Comparator, and Outcomes), defined as follows:Population/Participants (P): Studies including children and adolescents (≤18 years at treatment initiation) with any type of cancer or undergoing hematopoietic stem cell transplantation treated with chemotherapy, radiotherapy, or chemoradiotherapy, with follow-up data permitted up to 25 years of age.Intervention/Index (I): Nutritional supplementation, dietary modification, or widely available natural products (e.g., honey, omega-3 fatty acids, Aloe vera, olive oil, probiotics, glutamine, herbal preparations, or similar agents) administered orally/systemically, with the aim of preventing or treating oral mucositis and/or mucositis-associated pain.Comparator/Comparison (C): Standard of care, placebo, or alternative active interventions.Outcomes (O): Incidence, severity, and duration of oral mucositis; mucositis-associated pain; and, where reported, secondary outcomes such as treatment tolerability, nutritional status, or quality of life.

### 2.2. Search Strategy

A systematic literature search was conducted across three databases to maximize coverage, namely, Scopus, PubMed/MEDLINE, and Google Scholar. Searches covered a timeframe spanning 1 January 2000–1 June 2025 and were performed on 17 August 2025. This multiplatform approach was chosen to overcome the limitations of individual databases, such as advanced search restrictions or filtering limitations.

The search strategy combined controlled vocabulary (Medical Subject Headings [MeSH]) and free-text terms related to the population, condition, and interventions of interest. Boolean operators (AND, OR) were applied to refine the search results. An example of applying the search strategy, including exact query strings, is provided as follows: “TITLE-ABS-KEY (child* OR adolescent* OR pediatric* OR pediatric* OR “young patient*” OR teen*) AND TITLE-ABS-KEY (cancer OR oncolog* OR malignan* OR neoplas* OR leukemia OR lymphoma OR “stem cell transplant*” OR “bone marrow transplant*”) AND TITLE-ABS-KEY (“oral mucositis” OR mucositis OR stomatitis OR “oral complication*” OR “mouth sore*”)”.

Database-specific search methods were applied. In Scopus and Google Scholar, the search was structured with TITLE-ABS-KEY fields to capture terms in titles, abstracts, or keywords, whereas in PubMed, terms were searched within titles and abstracts ([tiab]) and supplemented by MeSH terms (e.g., “oral mucositis” [MeSH Terms]).

### 2.3. Selection of Articles

To ensure the quality and reliability of the included sources, two authors independently assessed the publications. Any disagreements were resolved through discussion or, when necessary, by consulting a third author. For the screening process, two independent reviewers (C-S.I. and R.M.H.) evaluated all records for eligibility. Inter-rater reliability measured by Cohen’s Kappa was ≈0.87, indicating a high level of agreement. Discrepancies were addressed through consensus or, if unresolved, by involving a third reviewer (N.-I.V.).

### 2.4. Inclusion Criteria

Articles were selected based on the following criteria:(1)Studies including children and adolescents (≤18 years, up to 25 years of age for follow-ups) with any type of cancer or undergoing hematopoietic stem cell transplantation treated with chemotherapy, radiotherapy, or chemoradiotherapy.(2)Studies evaluating nutritional supplementation, dietary modification, or widely available natural products (e.g., honey, omega-3 fatty acids, Aloe vera, olive oil, probiotics, glutamine, herbal preparations, or similar agents) administered for the prevention or treatment of oral mucositis and/or mucositis-associated pain.(3)Studies comparing the intervention against standard of care, placebo, or alternative active interventions.(4)Studies reporting clinical outcomes related to oral mucositis, including incidence, severity, duration, and associated pain.(5)Studies that assessed digestive tract mucositis, in which oral mucositis outcomes were clearly reported.(6)Studies involving human participants, specifically randomized controlled trials (RCTs). RCTs were chosen exclusively for inclusion because they provide the highest level of evidence, allow for proper comparison between interventions and controls, and ensure greater reliability and reproducibility of results.

Findings from animal or in vitro studies were considered only for context within the Introduction or Discussion.

For this review, oral mucositis was defined as inflammation and/or ulceration of the oral mucosa resulting from anticancer therapy, as reported by the authors of [33]. Studies assessing digestive tract mucositis were included only if oral mucositis outcomes were separately described, as mentioned in the above criteria.

### 2.5. Exclusion Criteria

The exclusion criteria consisted of the following:(1)Studies based on adult study populations or pediatric populations not receiving cancer therapy.(2)Non-RCT designs, including quasi-experimental studies, controlled clinical trials without randomization, observational cohorts, case–control studies, cross-sectional studies, case reports, case series, and preclinical (animal or in vitro) studies.(3)Articles published in languages other than English, without an available translation.(4)Publications not appearing in peer-reviewed journals.(5)Studies not yet published or lacking accessible full texts (abstract-only).(6)Publications with unsuitable formats, such as letters, case reports, editorials, conference abstracts, systematic reviews, or meta-analyses.

It is also worth noting that studies investigating chewing gum as a treatment or preventive measure for oral mucositis were not included, as chewing gum functions primarily through mechanical stimulation of salivary flow rather than through the delivery of nutrients or bioactive compounds. Consequently, it does not align with the nutritional and supplement-based scope of this review.

After finalizing the list of eligible studies, two independent reviewers (M.-I.S. and E.C.) extracted information using a standardized data collection checklist. Extracted variables included study identifier (first author and year), design, sample size, patient characteristics, intervention details, comparator(s), and reported outcomes.

Any missing or unclear information was noted as a limitation during critical appraisal. Studies were included in each synthesis according to the availability of relevant outcome data. Given the heterogeneity in reporting across studies, no data were imputed or statistically adjusted; analyses were conducted solely on the data as presented in the original publications.

An important exception is represented by two included studies, one by Uderzo et al. and the other by Treister et al. [36,37]. These studies were included because the analyzed data were from predominantly pediatric populations and the conclusions drawn were presented accordingly in the original publications (as can be seen from the titles of the articles). However, upon close inspection, one can notice the inclusion of a small number of young adult participants, which would technically meet the exclusion criteria. Considering the large study populations and the reduced impact of the adult participants on the results, the studies were retained. This specific demographic aspect is noted for transparency, particularly in the context of excluding studies focused on adult populations.

### 2.6. Quality and Risk of Bias Assessment of the Studies

Study quality was assessed using the NIH Study Quality Assessment Tools (www.nhlbi.nih.gov/health-topics/study-quality-assessment-tools; accessed on 1 September 2025) by two independent reviewers (M.-I.S. and E.C.), with disagreements resolved by a third (N.-I.V.). Risk of bias was evaluated in the same manner after standardized training, following the Cochrane Collaboration’s tool [38] and the Revised Cochrane risk-of-bias tool for randomized trials (RoB 2). Risk-of-bias visualizations (traffic light and summary plots) were produced with the robvis R package (version 0.3.0, 2019; McGuinness & Higgins, 2021) [39]. Full study ratings and plots are provided in the Appendix A.

## 3. Results

The initial search across Scopus, PubMed/MEDLINE, and Google Scholar identified a total of 5870 records (Scopus = 4715; PubMed = 1046; Google Scholar = 109). Searches combined controlled vocabulary and free-text terms related to pediatric populations, cancer treatment, and oral mucositis.

After applying time, language, and publication-type restrictions, the results were reduced to 3380 records across the three databases, which were then retained for deduplication and screening. After duplicate removal, 3073 unique records remained for screening. Title/abstract screening excluded 3012 records for being off-topic (e.g., not pediatric cohorts or not evaluating oral mucositis), for assessing non-nutritional/non-supplement interventions (e.g., chlorhexidine/benzydamine/palifermin rinses, low-level laser therapy, oral-care protocols, or chewing gum), or for not being primary clinical studies. A total of 61 articles proceeded to full-text assessment; one could not be retrieved due to persistent access failure, leaving 60 full texts appraised. Of these, 46 were excluded (adult-only populations, ineligible study design, absence of oral-mucositis outcomes, or lack of a nutritional/supplement exposure), yielding 14 eligible studies from the database search. Complementary citation chasing/snowballing, targeted website searches, and peer review suggestions identified 35 additional records; 1 lacked an English full text and was not assessed, 6 were duplicates of already-selected records, and 21 were excluded for adult-only populations or irrelevant exposures/outcomes, leaving 7 additional eligible studies. In total, 21 randomized controlled trials were included in the review, as detailed in the PRISMA flowchart diagram (Figure 1).

A total of 21 randomized controlled trials met the eligibility criteria and were included in this review, encompassing a total of 1478 patients. These studies span over two decades (2001–2023) and represent diverse geographic regions, cancer types, and treatment settings. Interventions encompassed systemically administered supplements, topically applied nutritional agents, and nutrient-based oral rinses, with sample sizes ranging from fewer than 20 to more than 200 patients. Table 1 provides an overview of the main characteristics of the included studies, including study design, population, intervention and comparator arms, and outcome assessment tools.

### 3.1. Systemically Administered Nutrient Supplementation

Among the included studies, seven RCTs evaluated systemically administered(oral/enteral/parenteral) nutritional supplements—such as glutamine, vitamin E, pycnogenol, bovine colostrum, and zinc—in children receiving chemotherapy and/or undergoing HSCT [36,41,42,44,53,54,57,58]. While individual findings varied, the overall pattern indicates that systemic supplementation does not consistently reduce the incidence or severity of oral mucositis, with the only palpable evidence of benefit being for pycnogenol and bovine colostrum.

Overall, the results have been inconsistent, with most studies failing to demonstrate a clear preventive effect on mucositis incidence or severity, the evidence being summarized in Table 2.

Glutamine was the most frequently studied nutrient, with four RCTs evaluating different formulations. Aquino et al. [41] reported a nonsignificant trend toward lower average mucositis scores (*p* = 0.07), but glutamine significantly reduced the duration of morphine use (12.1 vs. 19.3 days, *p* = 0.01) and TPN use (17.3 vs. 27.3 days, *p* = 0.02). Similarly, Ward et al. [44] observed shorter fever duration with glutamine (5.7 vs. 12.9 days, *p* = 0.021) and nonsignificant reductions in severe OM (29% vs. 55%, *p* = 0.118) and sinusoidal obstruction syndrome (10% vs. 35%, *p* = 0.067). In contrast, Uderzo et al. [36] found no benefit of glutamine-enriched TPN compared to standard TPN, with OM incidence exceeding 90% in both arms and no significant differences in severity, analgesic use, or clinical outcomes. Widjaja et al. [54] was the only study which showed a significant reduction in OM incidence with oral glutamine (4.2% vs. 62.5%, *p* = 0.001; OR 0.026, 95% CI 0.003–0.228). Besides that, shorter hospitalization (7.7 vs. 12 days, *p* = 0.005) and lower treatment costs (USD 314 vs. 622) without adverse effects were noted.

Vitamin E supplementation was tested by El-Housseiny et al. [42], who directly compared topical versus systemic administration. Topical vitamin E resulted in complete healing in 80% of children (*p* < 0.001), while systemic vitamin E showed no significant benefit (*p* = 0.317), highlighting the superiority of local delivery.

Bovine colostrum was investigated by Rathe et al. [53], who found that peak OM severity was significantly lower in the colostrum group compared to the placebo group (*p* = 0.02), with sensitivity analyses of patient-reported outcomes confirming reduced severity (*p* = 0.009). However, no differences were observed in infection rates, fever days, or treatment delays.

Two recent trials evaluated honey, olive oil, and zinc. Badr et al. [57] showed that both honey and olive oil significantly reduced OM grades compared with the control at day 7 (*p* = 0.01), with honey providing the lowest pain scores (*p* = 0.00) and best acceptability. In contrast, Shah et al. [58] reported that zinc supplementation had no effect on OM incidence (20.5% vs. 19.6%, *p* = 0.91), severity (*p* = 0.79), onset, or duration.

Taken together, these studies suggest that while systemic supplementation with glutamine and zinc has not consistently reduced the incidence or severity of oral mucositis, glutamine may confer supportive benefits by reducing fever and hospitalization duration and need for parenteral nutrition and analgesia. In contrast, locally delivered nutritional agents such as topical vitamin E and honey show more robust and reproducible effects, with bovine colostrum offering preliminary evidence of benefit in attenuating mucositis severity [42,53,57].

### 3.2. Topically Applied Nutritional Agents

Nine randomized controlled trials evaluated the effect of topical nutritional agents applied directly to the oral mucosa in children with cancer [42,45,46,48,50,52,55,56,57]. The summarized findings can be found in Table 3.

Vitamin E has been investigated in two trials, with El-Housseiny et al. [42] showing that topical application resulted in complete healing in 80% of patients (*p* < 0.001), whereas systemic administration had no benefit (*p* = 0.317). Khurana et al. [46] further demonstrated that topical vitamin E and pycnogenol both significantly reduced OMAS and ChIMES scores compared to a control (*p* < 0.001), with complete healing in 75% and 58% of children, respectively; vitamin E remained effective even in grade 4 OM, whereas pycnogenol did not.

Honey-based interventions showed benefits in the most consistent manner compared to other findings. Abdulrhman et al. [45] reported faster recovery with honey compared to both a HOPE mixture and benzocaine gel, particularly for grade 3 OM (5.4 vs. 8.6 days; *p* < 0.01). Al Jaouni et al. [50] confirmed that honey significantly reduced the incidence of grade III–IV OM (20% vs. 55%; *p* = 0.02), lowered Candida and bacterial colonization (both 10% vs. 60%; *p* = 0.003), shortened hospital stay (7 vs. 13 days; *p* < 0.001), and improved weight gain. In the most recent trial, Badr et al. [57] found that honey and olive oil both reduced OM severity compared with a sodium bicarbonate control (*p* = 0.01), with honey providing the greatest pain relief (*p* = 0.00) and better acceptability than olive oil.

Olive oil and Aloe vera also demonstrated protective effects. In a small trial, Alkhouli et al. [52] showed that olive oil delayed OM onset (mean week 4.33 vs. 2.27; *p* = 0.022), reduced severity across weeks 2–8 (*p* < 0.01), and prevented grade 4 OM compared with sodium bicarbonate. In following studies, Aloe vera significantly delayed onset (week 4.3 vs. 2.3; *p* = 0.001) and lowered severity at several time points [55], while both Aloe vera and olive oil outperformed sodium bicarbonate in reducing OM severity, with no significant difference between the two [56].

Other natural products produced mixed findings. In a trial looking into benefits of Chinese propolis extract, no significant differences were found compared with a placebo (42% vs. 48% severe OM; *p* = 0.59) [48]. The HOPE mixture containing propolis, honey, olive oil, and beeswax also underperformed relative to honey alone, with higher rates of burning sensations [45].

In summary, these findings show that topical natural agents can be considered the most consistently effective nutritional approach in pediatric OM. The strongest evidence supports honey, which reproducibly reduces severity, pain, infection, and hospitalization. Vitamin E also shows clear benefits when applied topically, though not systemically. Olive oil and Aloe vera demonstrate promising results across multiple trials, while propolis alone does not show significant efficacy [45,48,52,55,56,57].

### 3.3. Nutrient-Containing Oral Rinses

Six randomized trials evaluated nutrient-based oral rinses for the prevention or management of oral mucositis in children, including Traumeel S, Caphosol (supersaturated calcium–phosphate solution), vitamin E rinse, and chamomile mouthwash [37,40,43,47,49,51]. These studies tested swish-and-spit or swish-and-swallow protocols administered multiple times daily, typically during chemotherapy or hematopoietic stem cell transplantation. Results have been largely inconsistent: early single-center data suggested potential benefits for Traumeel S, but this was not reproduced in a large multicenter RCT; both small and large trials of Caphosol found no efficacy and in some cases worse outcomes; vitamin E rinse did not demonstrate clinically relevant benefits, whereas chamomile mouthwash showed delayed but significant improvements after two weeks of use. The key characteristics and findings of these trials are summarized in Table 4.

These trials indicate that nutrient-containing oral rinses have not consistently demonstrated efficacy in pediatric mucositis. Initial single-center results with Traumeel S suggested delayed progression and lower mucositis scores [40], but a subsequent large multicenter RCT failed to confirm any benefit [47]. Similarly, Caphosol was tested in both a small exploratory study [40] and a large, rigorous, international trial [37], with both reporting no reduction in severe mucositis, with the drawback of lower compliance in younger patients. Vitamin E rinse also did not achieve clinically meaningful improvement over placebo [43]. By contrast, chamomile mouthwash showed a delayed but statistically significant reduction in mucositis severity at day 14 compared to an active comparator rinse [51].

Overall, the weight of evidence does not support routine use of nutrient-based rinses in children, with the exception of chamomile, which may warrant further study.

## 4. Discussion

### 4.1. General Considerations

This systematic review of 20 randomized controlled trials in children and adolescents with cancer found that topical nutritional agents, particularly honey and vitamin E, showed the most consistent benefits for reducing the severity and duration of oral mucositis, whereas systemic supplementation with glutamine, zinc, or colostrum yielded insufficient and inconclusive evidence.

While the pediatric population presents unique biological and clinical characteristics, valuable insights can be drawn from the considerably larger body of adult literature. In adults, particularly those with head and neck cancers undergoing chemoradiotherapy, oral mucositis is a major driver of malnutrition, treatment interruptions, and reduced quality of life [59]. Consequently, a wide range of nutritional and natural interventions have been tested, resulting in dozens of RCTs and several systematic reviews and meta-analyses. Recent high-quality reviews, including de Sousa Melo et al. (2021) and Lima et al. (2021), consistently report that natural products such as honey and Aloe vera show the most reproducible benefits, with additional though less consistent signals for agents like chamomile, curcumin, and certain polyphenol-rich extracts [60,61]. These interventions appear to reduce the severity and duration of mucositis, relieve pain, and in some cases improve quality of life, although heterogeneity in trial design, cancer types, and dosing strategies remains a challenge.

By comparison, our review highlights that pediatric evidence is much more limited, with fewer RCTs, smaller sample sizes, and a narrower range of tested interventions. Nonetheless, the patterns of evidence are supported and in concordance with the adult literature findings.

### 4.2. Most Impactful Nutritional Agents on Prevention and Evolution of Oral Mucositis

Across all three categories, several patterns emerge regarding the most promising nutrient-based interventions for pediatric oral mucositis. Honey stands out as the most consistently effective agent: in multiple RCTs it significantly reduced mucositis severity and recovery time [45], decreased rates of Candida and bacterial colonization [50], lowered pain scores and hospital stay [50], and outperformed olive oil in head-to-head testing [57]. Topical vitamin E also showed reproducible benefit, with El-Housseiny et al. [42] reporting complete healing in 80% of children and Khurana et al. [46] confirming significant reductions in mucositis and pain scores, including in severe cases. Olive oil and Aloe vera demonstrated consistent though somewhat smaller benefits, delaying onset and lowering severity compared to sodium bicarbonate [52,55], with results confirmed in follow-up research [56]. Bovine colostrum showed a protective signal in reducing peak mucositis severity [53], although its clinical impact beyond symptom grading was limited.

### 4.3. Importance of Oral Hygiene

Although many nutritional and natural products have shown promise, perhaps the most fundamental determinant of oral mucositis outcomes is the quality and consistency of basic oral hygiene practices. Recent pediatric studies demonstrate that implementing standardized oral care protocols—including regular toothbrushing, bland rinses, and caregiver/child education—can substantially reduce the incidence and severity of mucositis, even before the addition of any specific nutritional agent [29,62,63,64]. Nutritional supplements and topical applications may therefore be best viewed as adjuncts to, rather than replacements for, rigorous oral care, reinforcing the importance of integrating these measures into comprehensive supportive care protocols. This implies necessity for permanent programs that educate both the general population, but mostly risk population, on the importance of oral hygiene.

Across the literature and guidelines in pediatric oncology (supported by adult studies) the quality, consistency, and timing of oral hygiene strongly shape OM outcomes [65]. Standardized protocols (toothbrushing, bland rinses, lip care, caregiver/patient education, and early dental assessment) lower incidence, delay onset, and reduce severity, even before adding any specific agent [66,67,68].

There are guidelines for the prevention of oral and oropharyngeal mucositis in children receiving treatment for cancer or undergoing hematopoietic stem cell transplantation [60,69,70,71,72]. In pediatrics, an implementation project in HSCT cut OM incidence from 66.6% to 36.7% and eliminated grade III–IV cases after embedding routines and audited oral care workflows, underscoring the impact of process alone [62,73]. Pediatric systematic review data point the same way: integrated oral care—often 2–6×/day—is associated with lower severity, pain, and duration; the choice of agent then modulates the effect size [62].

### 4.4. Implications for Clinical Practice

Current MASCC/ISOO guidelines [67] emphasize the importance of basic oral care protocols and patient and caregiver education, as well as procedures such as cryotherapy and photobiomodulation (PBM). Pharmacological interventions such as palifermin are effective in selected high-risk settings but remain costly and are often inaccessible outside high-resource centers. These represent core evidence-based strategies for pediatric oral mucositis, while acknowledging the limited number and quality of pediatric RCTs and the need for extrapolation from adult data.

Our systematic review respects the importance of oral hygiene and recognizes the role of established procedures, as well as a need for pharmacological solutions for selected situations. However, it also highlights that several low-cost, natural, nutrition-based interventions have shown reproducible benefit in pediatric RCTs, with reductions in mucositis severity, faster healing, less pain, and even lower infection rates and shorter hospitalization.

Before detailed and discussed findings, we suggest potential clarifications to current guidance [28]:Adjunctive use of honey and other accessible natural products may be recommended in centers where PBM and palifermin are unavailable, particularly in resource-limited settings.Oral care remains foundational, and natural topical agents should be considered as adjuncts rather than replacements.Routine use of systemic supplementation or Caphosol rinses is not supported in pediatrics and may divert resources without clinical benefit.

In accordance with MASCC/ISOO guidelines [63,67,74], a pragmatic strategy for a lower-funding context may take the following form:

Tier 0 (no-cost/low-cost must-dos): Basic oral care bundle + caregiver education sheet + daily checklist; bland rinses; petroleum jelly for lips. Use popsicles for cryotherapy during short infusions. Track with ChIMES/WHO scores simplified for age.

Tier 1 (very low-cost adjuncts): Honey (medical-grade/pasteurized), topical vitamin E, Aloe vera 70%, olive oil swabs, chamomile rinse (standardized). Add one at a time, document OM grade/pain daily, and reassess at 48–72 h. (Apply caries prevention steps when using honey.)

Tier 2 (mid-resource): Extra-oral PBM using LED devices with pediatric-friendly application (better tolerance than intra-oral in some cohorts); follow conservative energy settings and protective measures per institutional protocol.

Tier 3 (high-resource): Intra-oral PBM by trained staff; palifermin for selected HSCT regimens with toxicity-reduction goals.

In practice, structured oral care remains the foundation, with topical nutritional agents used as adjuvants in current practice, rather than replacements. This approach is adaptable particularly resource-limited settings, where low-cost topicals can be incorporated into standardized oral-care combined practices, while high-resource options (e.g., PBM and palifermin) are used when available or when severity indicates a need for them.

### 4.5. Limitations

This review has several limitations that should be acknowledged. First, despite a comprehensive search and strict inclusion of randomized controlled trials, the overall evidence base in pediatric populations remains small, with most studies being single-center, underpowered, and heterogeneous in design. Sample sizes were typically modest, not only limiting statistical power but also forcing frequent reliance on adult data for context and comparison. While this approach is necessary to interpret findings in a broader clinical framework, it introduces a risk of indirectness and bias, since adult populations differ from children in ways which were previously discussed. As a result, extrapolating from adult evidence may overestimate or underestimate the true efficacy of nutritional interventions in pediatric oncology.

Second, there was substantial heterogeneity in terms of cancer diagnoses, treatment regimens, mucositis risk, intervention formulations (e.g., different types and concentrations of honey, vitamin E, or Aloe vera), and dosing schedules. This variability limited our ability to perform pooled quantitative analyses and increased the risk that observed benefits or null findings were context-dependent.

Third, blinding and placebo control were frequently challenging due to the taste, smell, or texture of natural products such as honey, olive oil, or vitamin E, raising the possibility of performance and detection bias. Compliance reporting was also variable and sometimes suboptimal, particularly for rinses.

Fourth, most studies focused on short-term outcomes (mucositis incidence, severity, and pain), while few assessed long-term effects such as nutritional status, growth, treatment adherence, or quality of life. Similarly, reporting of adverse effects was inconsistent, making it difficult to comprehensively assess safety. Also, there was a degree of heterogeneity in participant age concerning the inclusion of two studies (Treister et al. and Uderzo et al.) whose cohorts extended slightly beyond the strict pediatric age threshold (13.7 (4.0–21.9) years and 8.1 (0.4–18.6), respectively) [43,49].

Fifth, while our review was restricted to RCTs to ensure methodological rigor, this decision excluded some potentially relevant observational studies that may somewhat reflect real-world practice, especially in low-resource settings.

Finally, although we prospectively registered our review protocol and followed PRISMA guidance and rigorous methodology, the possibility of publication bias is worth mentioning.

### 4.6. Future Research

Future studies should focus on conducting large, multicenter pediatric RCTs to confirm the benefits of promising low-cost topical agents such as honey, vitamin E, olive oil, and Aloe vera, using standardized outcome measures (e.g., WHO, OMAS, and ChIMES) and incorporating patient-reported outcomes. Research should also examine long-term effects on nutrition, growth, quality of life, and treatment adherence, which are rarely reported. Comparative trials between natural agents and guideline-endorsed interventions such as photobiomodulation or palifermin would help clarify their relative value. Standardized methodology for treatments and efficacy evaluation is needed to provide context for more robust secondary data analysis.

Finally, implementation studies in resource-limited settings are needed to evaluate feasibility, acceptability, and cost-effectiveness, ensuring that evidence-based supportive care strategies are accessible across diverse healthcare contexts.

## 5. Conclusions

Evidence from pediatric RCTs suggests a hierarchy of benefits: topical, nutrition-based agents outperform systemic supplements and nutrient rinses for preventing or mitigating OM in children. Benefits are most consistent for honey and vitamin E, with supportive signals for Aloe vera, olive oil, and chamomile; by contrast, glutamine and zinc have not shown convincing enough evidence in reducing OM incidence or peak severity, and Caphosol/Traumeel lack reproducible efficacy.

Given the small and heterogeneous pediatric evidence base, definitive guidance awaits larger, multicenter trials using standardized, patient-centered outcomes and direct comparisons with established modalities.

This shows the importance of guideline implementation with regard to local resource availability and real-world clinical practice.

## Figures and Tables

**Figure 1 nutrients-17-03521-f001:**
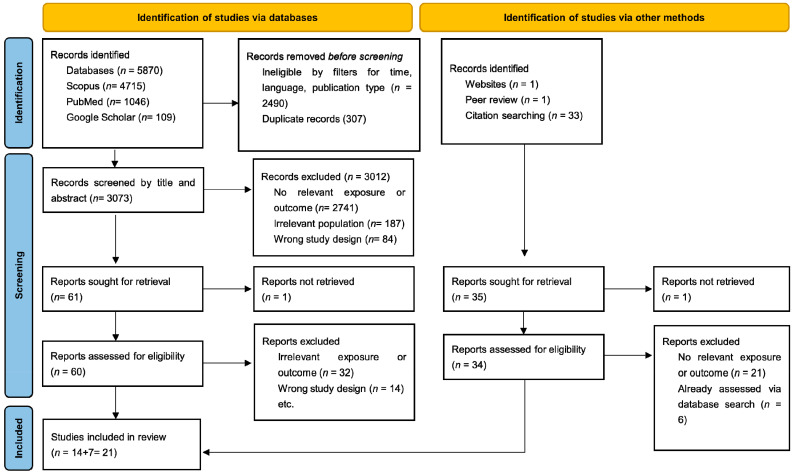
Study selection process PRISMA flowchart.

**Table 1 nutrients-17-03521-t001:** Overview table of the selected studies. NR: Not reported, WHO: World Health Organization (oral mucositis scale), HSCT: hematopoietic stem cell transplantation, post-SCT: post-stem cell transplantation, OM: oral mucositis, GE-TPN: glutamine-enriched total parenteral nutrition, S-TPN: standard total parenteral nutrition, HOPE: honey + olive oil–propolis extract + beeswax, ChIMES: Children’s International Mucositis Evaluation Scale, OMAS: Oral Mucositis Assessment Scale, OAG: Modified Oral Assessment Guide, CTC: Common Toxicity Criteria, VAS: Visual Analogue Scale, NCI-CTCAE: National Cancer Institute Common Terminology Criteria for Adverse Events, OMDQ: Oral Mucositis Daily Questionnaire, post-HCT: post-hematopoietic cell transplantation, TID: three times daily.

Author	Year	Country	Study Design	Number of Patients ^1^	Number of Patients in Intervention Group	Age (Median or Mean Age ± SD)	Tumor Type	Assessment Measures	Intervention Group	Control Group	Reference Number
Oberbaum	2001	USA	Randomized, double-blind, placebo-controlled clinical trial	30	15	9.9 ± 6.38 years	NR	WHO	Traumeel S oral rinse (homeopathic complex of 14 herbs/minerals), 5× daily from day + 2 post-SCT for ≥14 days or until OM resolution.	Placebo (saline solution, indistinguishable in appearance and taste)	[40]
Aquino	2005	USA	Randomized controlled study	120	57	9.81 ± 0.81 years	Mixed malignancies	Walsh	Oral glutamine at a dose of 2 g/m^2^/dose (maximum dose 4 g) administered in a solution of 500 mg/mL twice daily beginning on the day of admission for HSCT.	Oral glycine at a dose of 2 g/m^2^/dose (maximum dose 4 g) administered in a solution of 500 mg/mL twice daily beginning on the day of admission for HSCT.	[41]
El-Housseiny	2007	Egypt	Randomized controlled trial (two-arm)	80	80	5.75 + 3.38 years	Mixed malignancies	WHO	Group A (topical): 100 mg vitamin E (from 100 IU capsule) emptied into mouth twice daily × 5 days.	Group B (systemic): 100 mg vitamin E capsule swallowed twice daily × 5 days.	[42]
Sung	2007	Canada	Serial N-of-1 randomized, double-blind, placebo-controlled trials combined with Bayesian meta-analysis	16	16	12.7 years (range 6.4–15.1 years)	Pediatric cancers requiring ≥2 identical doxorubicin-containing chemotherapy cycles (Ewing’s sarcoma, lymphoma, osteosarcoma, rhabdomyosarcoma)	Objective mucositis scale, WHO mucositis scale (0–4), VAS pain/swallowing	Topical vitamin E solution (800 mg DL-α-tocopheryl acetate in corn oil, 2 mL once daily × 14 days after each doxorubicin-containing cycle, swish and spit)	Placebo solution (corn oil carrier, identical taste/appearance)	[43]
Ward	2009	UK	Randomized cross-over study (patients as own controls)	50	50	8.7 ± 5.8 years	Mixed malignancies	CTC	Enteral glutamine 0.65 g/kg/day (oral or via NG/gastrostomy) × 7 days during chemo	Identical chemotherapy course without glutamine (self-control design)	[44]
Uderzo	2011	Italy	Prospective multicenter randomized double-blind controlled trial	118	62	8.1 years (range 0.4–18.6)	Mixed malignancies	WHO	GE-TPN containing 0.4 g/kg/day L-alanyl-glutamine dipeptide (≈0.25 g/kg glutamine)	S-TPN without glutamine enrichment	[36]
Abdulrham	2012	Egypt	Randomized controlled trial	90	60	6.9 ± 3.8 years	ALL	NCI-CTC	Group 1: Honey (0.5 g/kg, max 15 g, applied topically TID × 10 days); Group 2: HOPE mixture (honey + olive oil–propolis extract + beeswax, 0.25 g/kg, max 5 g, topically TID × 10 days)	Benzocaine 7.5% gel, TID × 10 days	[45]
Khurana	2013	India	Randomized single-blind controlled trial (three-arm)	72	48	Group 1: 8.98 ± 2.58 years Group 2: 9.29 ± 2.58 years Group 3: 9.48 ± 2.53 years	Mixed malignancies	WHO/ChiMES/OMAS	Group II: Vitamin E (200 mg/day topical solution in glycerine, 3× daily × 7 days); Group III: Pycnogenol (pine bark extract, 1 mg/kg/day in glycerine, 3× daily × 7 days)	Group 1: Glycerine solution	[46]
Sencer	2012	USA and Israel	International multicenter double-blind randomized placebo-controlled trial	181	98	Intervention group: 12 years (3.24) Control group: 11 years (3.25)	Mixed malignancies	Modified Walsh mucositis scale (daily, days −1 to +20) and WHO scale	Traumeel S oral rinse (complex homeopathic solution with 14 herbal/mineral components), 5× daily from day −1 to day +20 after HSCT	Placebo saline solution (identical ampoules)	[47]
Tomaževič	2013	Slovenia	Double-blind randomized placebo-controlled trial	40	19	Intervention group: 6.7 ± 5.3 years Control group: 9.3 ± 6.6 years	Mixed malignancies	OAG, score 3 = severe OM	70% ethanolic extract of Chinese propolis (0.38 g per application, applied twice daily to vestibular mucosa)	Placebo solution (70% alcohol + caramel dye, matched for color, viscosity, taste)	[48]
Raphael	2014	The Netherlands	Randomized controlled trial	29	15	Intervention group: 11.3 ± 3.9 years Control group: 9.9 ± 4.7 years	Mixed malignancies	NCI-CTCAE	Caphosol (supersaturated calcium phosphate rinse), 4× daily during OM episode	Placebo (NaCl 0.9% rinse), identical in appearance and taste	[49]
Treister	2016	USA, Canada, Australia/NZ	Phase III international multicenter randomized double-blind placebo-controlled trial	220	110	13.7 (4.0–21.9) years	Mixed malignancies	WHO Oral Toxicity Scale (grades 0–4); also Mouth Pain Categorical Scale (0–10) and OMDQ	Caphosol oral rinse (supersaturated calcium phosphate electrolyte solution), 4× daily from start of conditioning to day +20 post-HCT	Placebo oral rinse (0.9% saline), same schedule	[37]
Al Jaouni	2017	India	Randomized controlled trial	40	20	Intervention group: 7.9 ± 4.1 years Control group: 8.1 ± 4.9 years	Mixed malignancies	WHO	Local Saudi commercial honey, applied topically 4–6× daily to oral mucosa + saline rinse	Standard oral hygiene (lidocaine, Mycostatin, Daktarin gel, mouthwash) without honey	[50]
Pourdeghatkar	2017	Iran	Randomized double-blind clinical trial	62	31	Intervention group: 9.9 ± 2.9 years Control group: 9.7 ± 3.01 years	ALL	WHO	Chamomile mouthwash (30 drops diluted in 20 mL water, swish/gargle 1 min, 3× daily for 14 days, starting 1 day before chemotherapy).	Topical mouth rinse (sucralfate, allopurinol, bicarbonate 7.5%, half-saline solution, 20 mL swish 3× daily × 14 days).	[51]
Alkhouli	2019	Syria	Triple-blind randomized controlled clinical trial	22	11	Intervention group: 5.4 years Control group: 5.2 years	ALL	WHO	Topical olive oil, swabbed twice daily on oral mucosa (tongue, buccal mucosa, lips, palate) starting 2 days before chemotherapy	Sodium bicarbonate 5% solution	[52]
Rathe	2020	Denmark	Randomized, double-blind, placebo-controlled trial	62	30	Intervention group: 4 years (1–15) Control group: 5 years (2–14)	ALL	NCI-CTCAE	Bovine colostrum powder (0.5–1 g/kg/day, reconstituted in water, oral/NG tube, daily × 29 days)	Isocaloric milk protein powder placebo	[53]
Widjaja	2020	Indonesia	Randomized, double-blind, placebo-controlled trial	48	24	Intervention group: 6.29 ± 4.42 years Control group: 5.9 ± 2.9 years	ALL	WHO	Oral glutamine at a dose of 400 mg/kg/day started 24 h before methotrexate for 14 days	Placebo not detailed	[54]
Alkhouli	2021	Syria	Triple-blind randomized controlled clinical trial	26	11	Intervention group: 4.6 years Control group: 4.8 years	ALL	WHO	70% Aloe vera solution, applied topically with sponge swab 2× daily starting 3 days before chemo, continued during induction	Sodium bicarbonate 5% (control)	[55]
Alkhouli	2021	Syria	Randomized controlled three-arm clinical trial (double-blind)	36	22	Intervention group A: 7.5 years Intervention group B: 8.1 years Placebo group: 6.9 years	ALL	WHO	Group A: Aloe vera 70% solution (swab, 4× daily × 10 days) Group B: Extra virgin olive oil (swab, 4× daily × 10 days)	Sodium bicarbonate 5% solution (swab, 4× daily × 10 days)	[56]
Badr	2023	Lebanon	Single-blind randomized controlled phase II trial (3 arms)	46	20	Intervention group 1: 10.89 ± 4.10 years Intervention group 2: 9.63 ± 4.17 years Control group: 9.03 ± 3.98 years	ALL	WHO	Group 1: Manuka honey (2.5 cc, swish 1 min then swallow, TID × 7 days) Group 2: Extra virgin olive oil (2.5 cc, swish 1 min then swallow, TID × 7 days)	Standard care (5 cc 3% sodium bicarbonate + 5 cc Rinsidin, swish and spit, TID × 7 days)	[57]
Shah	2023	India	Double-blind randomized placebo-controlled trial	90	44	Intervention group: 6 (4;11) years Control group: 7 (4;10) years	Mixed malignancies	WHO	Oral zinc gluconate syrup, 1 mg/kg/day (max 30 mg), once daily × 14 days.	Placebo syrup (matched for taste, color, and smell).	[58]

^1^ Only randomized patients were included.

**Table 2 nutrients-17-03521-t002:** Characteristics and outcomes of trials on ingested and systemic nutrient supplementation. HSCT: Hematopoietic stem cell transplantation, TPN: total parenteral nutrition, OM: oral mucositis, GE-TPN: glutamine-enriched total parenteral nutrition, S-TPN: standard total parenteral nutrition, VAS: Visual Analogue Scale, NS: not significant, GVHD: graft-versus-host disease, TRM: transplant-related mortality, TID: three times daily.

Author (Year)	Country	Population (*n*) ^1^	Intervention	Comparator	Key Findings	Reference
Aquino (2005)	USA	120	Oral glutamine at a dose of 2 g/m^2^/dose (maximum dose 4 g) administered in a solution of 500 mg/mL twice daily beginning on the day of admission for HSCT.	Oral glycine at a dose of 2 g/m^2^/dose (maximum dose 4 g) administered in a solution of 500 mg/mL twice daily beginning on the day of admission for HSCT.	A non-significant trend toward reduced average mucositis scores with glutamine (*p* = 0.07); no difference in maximum mucositis score (*p* = 0.7). Significant reduction in days of morphine use with glutamine (12.1 vs. 19.3 days, *p* = 0.01). Significant reduction in days of TPN use with glutamine (17.3 vs. 27.3 days, *p* = 0.02). No excess toxicity was observed with glutamine compared to glycine. No significant differences in bacteremia episodes (*p* = 0.9), total hospital days (*p* = 0.4), or day-100 mortality (*p* = 0.7).	[41]
El-Housseiny (2007)	Egypt	80	Group A: 100 mg vitamin E (from 100 IU capsule) emptied into mouth twice daily × 5 days. Group B: 100 mg vitamin E capsule swallowed twice daily × 5 days.	Head-to-head comparison between topical and systemic vitamin E.	Topical group: 80% (24/30) healed completely; significant improvement (*p* < 0.001). Systemic group: No complete healing, most remained at grade ≥ 1; no significant improvement (*p* = 0.317). Topical vitamin E was well-tolerated; systemic administration not effective at tested dose.	[42]
Ward (2009)	UK	50	Enteral glutamine 0.65 g/kg/day (oral or via NG/gastrostomy) × 7 days during chemotherapy	Identical chemotherapy course without glutamine (self-control design)	Fever: Duration significantly shorter with glutamine (5.7 vs. 12.9 days, *p* = 0.021). Infections: Lower (38% vs. 55%). Severe OM (grade 3–4): Lower with glutamine (29% vs. 55%), but NS (*p* = 0.118). SOS incidence: Lower in glutamine group (10% vs. 35%), borderline (*p* = 0.067). Drug-related toxicity: Lower with glutamine (14% vs. 40%, *p* = 0.085). Engraftment, GVHD, hospital stay, mortality were similar.	[44]
Uderzo (2011)	Italy	118	GE-TPN containing 0.4 g/kg/day L-alanyl-glutamine dipeptide (≈0.25 g/kg glutamine)	S-TPN without glutamine enrichment	Incidence of OM: 94.8% (S-TPN) vs. 96.7% (GE-TPN), *p* = 0.68. Severity: No significant differences in OM grade distribution (OR 1.73, 95% CI 0.27–11.27). Analgesic use: Duration and type of opioids/analgesics similar. Engraftment, infections, GVHD, TRM (8.6% vs. 11.7%), relapse (17.2% vs. 8.3%), and hospital stay all comparable. Weight, albumin, prealbumin, cholinesterase values unchanged between groups. No adverse effects or increased relapse risk with glutamine.	[36]
Rathe (2020)	Denmark	62	Bovine colostrum powder (0.5–1 g/kg/day, reconstituted in water, oral/NG tube, daily × 29 days)	Isocaloric milk protein powder placebo	No difference in fever days (median 0 in both groups). Oral mucositis: Peak severity lower with colostrum (*p* = 0.02); fewer severe cases (3% vs. 23% in placebo). Patient-reported OMDQ: Sensitivity analysis showed lower severity in colostrum group (*p* = 0.009). No differences in bacteraemia, IV antibiotics, or treatment delays.	[53]
Widjaja (2020)	Indonesia	48	Oral glutamine at a dose of 400 mg/kg/day started 24 h before methotrexate for 14 days	Placebo not detailed	Oral glutamine (400 mg/kg/day) significantly reduced oral mucositis in children with ALL receiving high-dose methotrexate, with an incidence of 4.2% vs. 62.5% for placebo (*p* = 0.001; OR 0.026, 95% CI 0.003–0.228). No severe (grade 3–4) cases occurred in the glutamine group. Hospital stay was shorter (7.7 vs. 12 days; *p* = 0.005). No adverse effects were reported.	[54]
Badr (2023)	Lebanon	46	Group 1: Manuka honey (2.5 cc, swish 1 min then swallow, TID × 7 days) Group 2: Extra virgin olive oil (2.5 cc, swish 1 min then swallow, TID × 7 days)	Standard care (5 cc 3% sodium bicarbonate + 5 cc Rinsidin, swish and spit, TID × 7 days)	On day 7, both honey and olive oil groups had significantly lower OM grades vs. control (*p* = 0.01). Honey group had the lowest pain scores on VAS (*p* = 0.00), superior to both olive oil and control. Olive oil reduced pain compared to control but less than honey. Children tolerated honey better than olive oil (taste acceptance issue for olive oil)	[57]
Shah (2023)	India	90	Oral zinc gluconate syrup, 1 mg/kg/day (max 30 mg), once daily × 14 days.	Placebo syrup (matched for taste, color, and smell)	Incidence of OM: 20.5% (zinc) vs. 19.6% (placebo), *p* = 0.91. Severity: No significant differences between groups (*p* = 0.79). Onset: Slightly delayed in zinc (5.2 days) vs. placebo (3.8 days), not significant (*p* = 0.09). Duration: 5.6 vs. 7.1 days (*p* = 0.18). Hospitalization: 6.8% vs. 8.7%. Well-tolerated, no adverse effects linked to zinc.	[58]

^1^ Only randomized patients were included.

**Table 3 nutrients-17-03521-t003:** Characteristics and outcomes of trials on topically applied nutritional agents. HOPE: honey + olive oil–propolis extract + beeswax, OM: oral mucositis, OMAS: Oral Mucositis Assessment Scale, ChIMES: Children’s International Mucositis Evaluation Scale, ARR: Absolute Risk Reduction, NNT: Number Needed to Treat, VAS: Visual Analogue Scale, TID: three times daily.

Author (Year)	Country	Population (*n*) ^1^	Intervention	Comparator	Key Findings	Reference
El-Housseiny (2007)	Egypt	80	Group A: 100 mg vitamin E (from 100 IU capsule) emptied into mouth twice daily × 5 days. Group B: 100 mg vitamin E capsule swallowed twice daily × 5 days.	Head-to-head comparison between topical and systemic vitamin E.	Topical group: 80% (24/30) healed completely; significant improvement (*p* < 0.001). Systemic group: No complete healing, most remained at grade ≥ 1; no significant improvement (*p* = 0.317). Topical vitamin E was well-tolerated; systemic administration not effective at tested dose.	[42]
Abdulrhman (2012)	Egypt	90	Group 1: Honey (0.5 g/kg, max 15 g, applied topically TID × 10 days); Group 2: HOPE mixture (honey + olive oil–propolis extract + beeswax, 0.25 g/kg, max 5 g, topically TID × 10 days)	Benzocaine 7.5% gel, TID × 10 days	Grade 2 mucositis: Recovery time significantly shorter with honey (3.6 days) vs. HOPE (4.2 days) and control (4.6 days) (*p* < 0.05). Grade 3 mucositis: Recovery time honey (5.4 days) vs. HOPE (5.8 days) faster than control (8.6 days, *p* < 0.01). Honey healed faster than both HOPE and control (*p* < 0.01). Adverse effects: HOPE caused transient burning in 27% due to propolis; honey was well-tolerated.	[45]
Khurana (2013)	India	72	Group II: Vitamin E (200 mg/day topical solution in glycerine, 3× daily × 7 days) Group III: Pycnogenol (pine bark extract, 1 mg/kg/day in glycerine, 3× daily × 7 days)	Group 1: Glycerine solution	Complete healing in 75% Vit E, 58.3% Pycnogenol, vs. 4.2% control (*p* < 0.001). OMAS: Significant reduction in scores in Vit E and Pycnogenol vs. control (*p* < 0.001); no difference between Vit E vs. Pycnogenol. ChIMES: Pain reduction significant in Vit E and Pycnogenol vs. control (*p* < 0.01 from day 4 onwards). Severe OM (grade 4): Pycnogenol less effective (no difference from control), while Vit E showed significant benefit (*p* = 0.006). Both agents were well-tolerated; isolated vomiting episodes likely chemo-related.	[46]
Tomaževič (2013)	Slovenia	40	70% ethanolic extract of Chinese propolis (0.38 g per application, applied twice daily to vestibular mucosa)	Placebo solution (70% alcohol + caramel dye, matched for color, viscosity, taste)	Severe OM: 42% (propolis) vs. 48% (placebo). No significant difference in frequency, duration, or severity of severe OM (*p* = 0.59).	[48]
Al Jaouni (2017)	India	40	Local Saudi commercial honey, applied topically 4–6× daily to oral mucosa + saline rinse	Standard oral hygiene (lidocaine, Mycostatin, Daktarin gel, mouthwash) without honey	Grade III–IV OM: 20% honey vs. 55% control (ARR 35%, NNT = 2, *p* = 0.02). Candida colonization: 10% honey vs. 60% control (ARR 50%, NNT = 2, *p* = 0.003). Aerobic bacterial infection: 10% honey vs. 60% control (ARR 50%, NNT = 2, *p* = 0.003). Hospital stay (per OM episode): 7 days (honey) vs. 13 days (control), *p* < 0.001. Body weight: Mean gain 35% (honey) vs. 15% (control), *p* < 0.001. Pain: Delayed onset, reduced severity with honey.	[50]
Alkhouli (2019)	Syria	22	Topical olive oil, swabbed twice daily on oral mucosa (tongue, buccal mucosa, lips, palate) starting 2 days before chemotherapy	Sodium bicarbonate 5% solution (swab, 4× daily × 10 days)	OM developed in 3/11 olive oil vs. 11/11 sodium bicarbonate patients. Olive oil group: onset of OM significantly later (mean week 4.33 vs. 2.27; *p* = 0.022). Severity: OM grades significantly less severe in olive oil group from week 2 through week 8 (*p* < 0.01). No cases of grade 4 OM in olive oil group.	[52]
Alkhouli (2021)	Syria	26	70% Aloe vera solution, applied topically with sponge swab 2× daily starting 3 days before chemo, continued during induction	Sodium bicarbonate 5% (swab, 4× daily × 10 days)	Aloe vera group developed OM later (mean week 4.3) vs. sodium bicarbonate (week 2.3), *p* = 0.001. Lower severity of OM in Aloe vera group at weeks 2, 3, 4, and 7 (*p* < 0.05). No differences at weeks 1, 5, 6, 8.	[55]
Alkhouli (2021)	Syria	36	Group A: Aloe vera 70% solution (swab, 4× daily × 10 days) Group B: Extra virgin olive oil (swab, 4× daily × 10 days)	Sodium bicarbonate 5% solution (swab, 4× daily × 10 days)	Aloe vera and olive oil groups both showed significant OM improvement (*p* = 0.007, *p* = 0.002). Sodium bicarbonate showed no improvement (*p* = 0.414). No significant difference between Aloe vera vs. olive oil, or Aloe vera vs. Sodium bicarbonate. Olive oil significantly better than sodium bicarbonate (*p* < 0.05).	[56]
Badr (2023)	Lebanon	46	Group 1: Manuka honey (2.5 cc, swish 1 min then swallow, TID × 7 days) Group 2: Extra virgin olive oil (2.5 cc, swish 1 min then swallow, TID × 7 days)	Standard care (5 cc 3% sodium bicarbonate + 5 cc Rinsidin, swish and spit, TID × 7 days)	On day 7, both honey and olive oil groups had significantly lower OM grades vs. control (*p* = 0.01). Honey group had the lowest pain scores on VAS (*p* = 0.00), superior to both olive oil and control. Olive oil reduced pain compared to control but less than honey. Children tolerated honey better than olive oil (taste acceptance issue for olive oil)	[57]

^1^ Only randomized patients were included.

**Table 4 nutrients-17-03521-t004:** Characteristics and outcomes of trials on nutrient-containing oral rinses. OM: Oral mucositis, WHO: World Health Organization (oral mucositis scale), NS: not significant, GI: gastrointestinal, VOD: veno-occlusive disease.

Author (Year)	Country	Population (*n*) ^1^	Intervention	Comparator	Key Findings	Reference
Oberbaum (2001)	Israel	30	Traumeel S oral rinse (homeopathic complex of 14 herbs/minerals), 5× daily from day +2 post-SCT for ≥14 days or until OM resolution	Placebo (saline solution, indistinguishable in appearance and taste)	Incidence: 33% Traumeel group did not develop OM vs. 7% placebo. Worsening of OM: 47% Traumeel vs. 93% placebo (*p* < 0.01). Mean AUC mucositis score: 10.4 (Traumeel) vs. 24.3 (Placebo), *p* < 0.01. Time to worsening: Significantly delayed with Traumeel (*p* < 0.001). Subjective symptoms (pain, dryness, dysphagia): Lower in Traumeel group. Safety: Well-tolerated; nausea led 2 patients to discontinue after one dose.	[40]
Sung (2007)	Canada	16	Topical vitamin E solution (800 mg DL-α-tocopheryl acetate in corn oil, 2 mL once daily × 14 days after each doxorubicin-containing cycle, swish and spit)	Placebo solution (corn oil carrier, identical taste/appearance)	Primary endpoint (objective mucositis score): Mean 0.2 (Vit E) vs. 0.3 (Placebo), ratio 0.90 (95% CR 0.57–1.37), probability Vit E better = 73% (not meeting pre-defined efficacy threshold >95%). Clinically significant reduction (>20%): Only 35% probability with vitamin E. Secondary outcomes: No significant differences in WHO mucositis scores, pain, swallowing difficulty, or opioid/hydration/TPN requirements. Compliance: Lower in vitamin E group (79% vs. 89%). Safety: No unexpected toxicities; oily texture led to acceptability issues.	[43]
Sencer (2012)	USA and Israel	181	Traumeel S oral rinse (complex homeopathic solution with 14 herbal/mineral components), 5× daily from day –1 to day +20 after HSCT	Placebo saline solution (identical ampoules)	Primary endpoint (AUC Walsh score): No significant difference (76.7 Traumeel vs. 67.3 Placebo, *p* = 0.13). WHO OM scores: No significant difference (AUC 24.4 Traumeel vs. 21.6 Placebo, *p* = 0.24). Narcotic use: Trend lower with Traumeel (17.7 vs. 28.5 mg/kg morphine equivalents, *p* = 0.2). TPN days: Similar (15.3 vs. 15.2). NG feeding: Similar (11 vs. 9 patients). Adverse events: Similar across groups (GI, cardiac, infection, bleeding, GVHD, VOD). Mortality (30 days): 17% Traumeel vs. 14% Placebo (NS). Compliance: Variable; many centers had low adherence, but subgroup analyses still showed no benefit.	[47]
Raphael (2014)	The Neatherlands	29	Caphosol (supersaturated calcium phosphate rinse), 4× daily during OM episode	Placebo (NaCl 0.9% rinse), identical in appearance and taste	Days with OM > grade 1: 9.9 (Caphosol) vs. 6.4 (Placebo), *p* = 0.154. Total OM duration: Trend longer in Caphosol group (15.8 vs. 10.2 days). Pain: More days with pain in Caphosol (11.3 vs. 7.3, *p* = 0.043). Analgesic use: Longer in Caphosol group (15.5 vs. 9.1 days, *p* = 0.035). Other outcomes (tube/TPN feeding, blood cultures, morphine use): No significant differences.	[49]
Treister (2016)	USA, Canada, Australia/NZ	220	Caphosol oral rinse (supersaturated calcium phosphate electrolyte solution), 4× daily from start of conditioning to day +20 post-HCT	Placebo oral rinse (0.9% saline), same schedule	Primary endpoint (duration of severe OM, WHO grade 3–4): No difference (4.5 days both arms, *p* = 0.99). Incidence severe OM: 63% Caphosol vs. 68% placebo (*p* = 0.44). Pain/OMDQ scores: No differences between groups. Opioid use: Similar incidence, dose, and duration. TPN: Required in 72% Caphosol vs. 78% placebo (*p* = 0.30); mean duration 11.4 vs. 13.6 days. Infections: Febrile neutropenia, invasive bacterial infection similar. Adverse events: None attributed to rinses; 3 deaths unrelated to study drug. Compliance: 72% completed ≥2 rinses/day; compliance lower in younger children.	[37]
Pourdeghatkar (2017)	Iran	62	Chamomile mouthwash (30 drops diluted in 20 mL water, swish/gargle 1 min, 3× daily for 14 days, starting 1 day before chemotherapy)	Topical mouth rinse (sucralfate, allopurinol, bicarbonate 7.5%, half-saline solution, 20 mL swish 3× daily × 14 days).	Day 7: No significant difference in OM severity between groups (*p* = 0.46). Day 14: Chamomile group had significantly lower OM severity vs. topical rinse group (Z = 3.23, *p* = 0.001). Overall, chamomile mouthwash was more effective than the comparator in reducing OM after 2 weeks. Both interventions were safe and tolerated.	[51]

^1^ Only randomized patients were included.

## Data Availability

The original contributions presented in this study are included in the article. Further inquiries can be directed to the corresponding author.

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
