# Peer review of "Nutritional and Supplemental Interventions for Prevention and Treatment of Oral Mucositis in Pediatric Oncology"

_nutrients, 2025, doi:10.3390/nu17223521_

Round 1
Reviewer 1 Report
Comments and Suggestions for Authors
Reduce and modify Abstract Section
Reduce Introduction Section, state the Aim briefly and clearly (some Refs are missing, why some
sentences in bold??)
In (Materials and) Methods Section some Refs are missing, eg. Inclusion/Exclusion Criteria,
K-index)
No sub-Sections in Discussion Section
Some Refs are missing
Why in bold??? (some sentences)
Remove the Section "Future Research"
Reduce Conclusion(s) Section and state the main outcomes only
Some Refs must be corrected

Author Response
Reviewer 1
Thank you for taking the time and effort to review our paper. Serious consideration was taken in the process of accounting and applying the suggestion you made.
Comments 1: Reduce and modify Abstract Section
Response 1: Thank you for this suggestion. We have reduced and modified the Abstract section. Now the abstract section follows the PRISMA 2020 for Abstracts checklist (10.1136/bmj.n71). The PRISMA 2020 for Abstracts checklist has been attached in the supplementary material section.
Comments 2: Reduce Introduction Section, state the Aim briefly and clearly (some Refs are missing, why some sentences in bold??)
Response 2:
We acknowledge that certain sections, particularly the detailed description of oral mucositis pathophysiology, which no longer exemplifies the involved molecules. This part of the Introduction can be condensed to a greater extent if so considered. However, we believe that much of the current content is usefull for providing a comprehensive context about: the biological mechanisms underlying the pathology, the distinctive characteristics of the paediatric population, and the rationale for focusing on nutritional and natural interventions as well as clearly defining them. These steps build a framework in which the reader can understand the scope and clinical relevance of the review.
The aims of the study were stated toward the end of the Introduction and further, defining the target population, intervention types, and study objectives. We are open to providing an even more concise statement, now present at the end of the introduction section.
Comments 3: In (Materials and) Methods Section some Refs are missing, eg. Inclusion/Exclusion Criteria, K-index)
Response 3: Extensively answered bellow when addressing the comments presented in the PDF manuscript
Comments 4: No sub-Sections in Discussion Section
Response 4: Same as Response 3
Comments 5: Some Refs are missing
Response 5: Addressed
Comments 6: Why in bold??? (some sentences)
Response 6: Only key-words of some parts were bolded due to emphasis. Has been addressed in order to be homogenous with the rest of the paper.
Comments 7: Remove the Section "Future Research"
Response 7: As answer bellow, future research represent a classic part of the Discussions section.
Comments 8: Reduce Conclusion(s) Section and state the main outcomes only
Response 8: Addressed.
Comments 9: Some Refs must be corrected
Response 9: Extensive response regarding specific references can be found bellow, with some agreement and resulting in provided citations, while other specific references simply seem unnecessary. However, if this comment refers to some references being misplaced or wrongly citated is unclear.
Comments present in the PDF file:
Comments 1: “Why not (and) a meta-analysis????” – referring to the study type
Response 1: A systematic review can be called a meta-analysis only when it includes a quantitative statistical synthesis of the results from multiple studies.
While the purpose of a Systematic Review is that of comprehensive and systematic summarization of relevant evidence on a specific objective, the Meta-Analysis statistically combines numerical data from comparable studies to produce secondary data analysis and pooled effect estimates. Meta-Analyses require sufficiently homogenous outcome data can be statistically aggregated. This is just not the case with the objective and context of out systematic review.
Comments 2: “Remove that Section”- referring to the Simple Summary section
Response 2: Simple Summary is present simply because previous publishing experiences had us compose one, and here we pre-emptively provided one. We can remove it if the editorial team considers it redundant
Comments 3: “Replace by: “/Aim”…..”- referring to the Background/Objectives subsection of the Abstract.
Response 3: While we have split this subsection into the 2 corresponding subsections, we did not replace Objectives with Aim as this wording is used in the standard Nutrients Template Word file.
Comments 4: “Materials and …”- referring to the Methods subsection of the Abstract.
Response 4: Again, this wording is used in the standard Nutrients Template Word file.
Comments 5: “Remove the time period” – referring to the search time frame
Comments 5: Not only this time period basically represents one of the inclusion criteria, upon consulting PRISMA Abstract guidelines we are reminded that the date of search must also be mentioned and can now be found present.
Comments 6: “Correct the spelling….” – it is unclear to which spelling this comment refers to
Response 6: The newer version of the Abstract now longer present the sentence in question.
Comments 7: “Refs are missing (66-78)”
Response 7: Thank you for pointing this aspect. Initially we did not provide specific citations as we considered the general aspects regarding OM incidence in chemotherapy to be established facts. We have addressed the aspect an provided corresponding citations.
Comments 8: “Such as??? (Refs)”
Response 8: Thank you for observing the lack of citing specific literature. Has been addressed.
Comments 9: “State Refs (128-142)”
Response 9: Former rows 128-142 contained the following: Regarding nutritional interventions against oral mucositis, they can be grouped into three categories: systemically administered supplements—intended to modulate host me-tabolism, immunity and inflammatory response from within; topical applications—that act directly at the site of mucosal injury to exert prolonged local antioxidant, an-ti-inflammatory, and wound-healing effects; and nutrient-containing mouth rins-es—designed to deliver antibacterial, antioxidant or anti-inflammatory substrates while also mechanically removing debris and potential harmful bacteria, with maximum cover-age.
The objective of this review is to systematically identify and synthesize available evi-dence on nutritional and supplement-based interventions for the prevention or treatment of oral mucositis in children and adolescents (≤18 years, including studies that covered young adult patients who initiated treatment in childhood and were included in paediatric cohorts up to 25 years of age due to follow-up period) with cancer or undergoing hemato-poietic stem cell transplantation. Specifically, the scope will include both dietary and top-ical nutritional supplements used for the prevention or reduction of oral mucositis in pae-diatric oncology.
First paragraph is just a neutral way to structure the intervention possibilities. It doesn’t assert efficacy—only modality and intent—so it can stand without references. Furthermore, “to exert prolonged local antioxidant, an-ti-inflammatory” has been nuanced to “intended to deliver prolonged local antioxidant, anti-inflammatory, and wound-healing effects” in order to show we are referring to neutral, generally accepted concepts which do not require specific citations.
Second Paragraph is purely procedural (population, intervention) and does not require citations of any sort.
Comments 10: “Why in bold?”
Response 10: We agree with this observation. To explain, this was made in order to highlight the types of intervention present in the studies, which at the time that paragraph was written seemed of particular importance. We have addressed this aspect.
Comments 11: “Reduce that paragraph, and state the Aim, briefly and clearly”
Response 11: A very brief paragraph stating the aims of the study is now present at the end of the introduction.
Comments 12: “Materials and…”
Response 12: In literature it is often the chosen to use the “Methods” heading in the context of Systematic Reviews since there are no actual materials involved. (in the classical sense of laboratory research, just the use of online data bases, just to give an example in close related field - Forne et al. [doi.org/10.1016/j.oraloncology.2023.106488]). We have modified in accordance with your suggestion.
Comments 13: “Refs (164-179)??????”
Response 13: PICO framework does not require a specific reference, and it is not custom (unlike PROSPERO or PRISMA) as it is the most used model for structuring clinical questions. When discussing PICO itself, such as the origin, arguing for the use or against it, and other such situations, then it is appropriate to cite Richardson et al. 1995 [PMID: 7582737]. An example of a Systematic Review which does not cite any PICO source, despite using it is that published by the European Association of Urology, Sakalis et al. 2022 [doi.org/10.1016/j.euros.2022.04.002]
Comments 14: “According to which mathematical type???(Refs)…”
Response 14: We believe the answer you are looking for is the following: According to Jacob Cohen's 1960 coefficient of agreement. However, it is standard practice to use Cohen’s Kappa coefficient when referring to the agreement rate of study selection by 2 independent reviewers. Bellow is the contingency table including all the studies (both through database search and citation search) and the calculation of Kappa coefficient.
|
Reviewer B: Include |
Reviewer B: Exclude |
Row Total |
|
|
Reviewer A: Include |
72 (a) |
8 (b) |
80 |
|
Reviewer A: Exclude |
13 (c) |
3,014 (d) |
3,027 |
|
Column Total |
85 |
3,022 |
3,107 |
We correctly rounded up the Kappa coefficient in the manuscript since it better describes the result.
Comments 15: “State Refs”
Response 15: The above comment being present in at row 209 of the first version of the manuscript, corresponding to the inclusion criteria we have defined we are unsure what is supposed to be referenced.
Comments 16: “State Refs”
Response 16: We have provided a reference, however we did not see it necessary to reference a source for articulating a definition of oral mucositis, in the context of said condition being detailed with numerous reference in the introduction.
Comments 17: “State Refs”
Response 17: Similar to comment 15, the above comment being present in at row 236 of the first version of the manuscript, corresponding to the exclusion criteria defined by us for the purpose of this article, therefore we are unsure what is supposed to be referenced.
Comments 18: “State Refs (248-252)”
Response 18: Paragraph refers to studies not included in the systematic review for stated reasons. We believe citing the is not only unnecessary but unrecommended.
Comments 19: “61”
Response 19: Thank you for pointing this out. We have addressed the different writing style.
Comments 20: “No sub-Section in that Section”
Response 20: The approach of subsections in Discussion section was used in order to structure the aspects in a more accessible manner for the reader. There are other Systematic Reviews in literature using a similar approach such as previously cited paper - European Association of Urology, Sakalis et al. 2022 [doi.org/10.1016/j.euros.2022.04.002].
Comments 21: “Remove the YEAR(S)”
Response 21: The years were placed in the text in parentheses not with regard to respective citation but to provide temporal context since the phrase utilises the word “Recent”. Therefore, year provide a specific timeframe for what recent is in the context of the presented information.
Comments 22: “Why in bold????”
Response 22: Thank you for pointing this out. The respective use of bold font puts emphasis on certain keywords and aspects. This is the result of common work and stylistic approach of the authors which contributed to the paper. As it is not consistent with the rest of the paper and as you have pointed out may be seen as redundant it has been removed.
Comments 23: “State Refs (514-520)”
Response 23: While the referred rows represent our own proposal for guidance revision and we do not see what paper should be cited, we have noticed that in the former rows 512-514 we are referring to “we suggest potential clarifications to current guidance”. Therefor we provided reference what the respective guidance is.
Comments 24: “Irrelevant data (570-580)”
Response 24: Future research subsection is a standard practice subject of discussion in the majority of published Systematic Reviews, if not studies in general. While a specific universally guideline that supports that cannot be cited to our knowledge, we can refer to guidelines such as Dunton, R. (2021). Discussion section for Research papers. San Jose: San Jose State University Writing Center.
Comments 25: “Reduce, and state the main outcomes only….”
Response 25: Conclusions section has been reduced. Main outcomes are present in the first and most consistent paragraph of the updated Conclusions section.
Reviewer 2 Report
Comments and Suggestions for Authors
This is a study assessing the effectiveness of nutritional and supplemental treatment to reduce oral mucositis in children treated for malignancies. The evidence of the various treatments is sparse, this is well discussed. A have the following comments:
What is meant by ‘excepting studies that covered young adult patients who initiated treatment in childhood and were included in paediatric cohorts up to 25 years of age’ that these studies were excluded (that is correct) or that they were included (incorrect)? We want to see the effects in a paediatric population, and not in youngsters 18-25 years in whom the treatment was started before the age of 18, but the main effect could be studied after 18 years of age. This is unclear as you later write at the inclusion criteria ‘≤18 years, up to 25 years of age for follow-ups’.
Why gave the initial search so many different results ‘ Scopus = 4715; PubMed = 1046; Google Scholar = 109’. It is a bit strange that in Scopus were found more than triple the amount found in Pubmed and even 47 times more than in Google scolar. What is your explanation?
The study of Uderzo also included patients >18 years. Could these patients be removed from their data, otherwise this study should be omitted.
The study of Treister also included patients >18 years. Could these patients be removed from their data, otherwise this study should be omitted.
Table 2, the study of Uderzo should be omitted unless separate data are obtained for patients <18 years.
Table 4, the study of Treister should be omitted unless separate data are obtained for patients <18 years.
The limitations of this study are well described!
Author Response
We appreciate the time and effort you have put into reviewing our work. We have tried to answer and accommodate your requests as good as possible.
Comments 1: What is meant by ‘excepting studies that covered young adult patients who initiated treatment in childhood and were included in paediatric cohorts up to 25 years of age’ that these studies were excluded (that is correct) or that they were included (incorrect)? We want to see the effects in a paediatric population, and not in youngsters 18-25 years in whom the treatment was started before the age of 18, but the main effect could be studied after 18 years of age. This is unclear as you later write at the inclusion criteria ‘≤18 years, up to 25 years of age for follow-ups’.
Response 1: Thank you for your feedback and for highlighting the need for clarity in our inclusion criteria. To address your concern, the topic in question was revised both in PICO and Inclusion Criteria section and we hope the current form better reflects our intent. To address any concerns the high end of the age range was 21.9 years of age, with a median age 13.7 years in the study published by Treister et al. [10.1038/bjc.2016.380]
Comments 2: Why gave the initial search so many different results ‘ Scopus = 4715; PubMed = 1046; Google Scholar = 109’. It is a bit strange that in Scopus were found more than triple the amount found in Pubmed and even 47 times more than in Google scolar. What is your explanation?
Response 2: The search comprised keywords that narrow the search on the subject at hand. The query for Scopus and Google Scholar were identical while the query for PubMed suffered technical changes to better suit the search engine. Possibly, better than any explanation we can provide the search queries along with the search results counts which were noted at the moment of the initial search. Not only that but I can also provide the counts at the current date (23 October 2025) which stand as follows: Scopus= 4,743 documents found, PubMed= 1,054 results, Google Scholar=116
Comments 3: The study of Uderzo also included patients >18 years. Could these patients be removed from their data, otherwise this study should be omitted.
The study of Treister also included patients >18 years. Could these patients be removed from their data, otherwise this study should be omitted.
Response 3: We acknowledge that these studies include a small number of patients >18 years. Unfortunately, the published data do not allow for the separation of outcomes specifically for patients ≤18 years. However, both studies predominantly involve paediatric cohorts and provide valuable data due to their large sample sizes and comprehensive outcomes relevant to our research question. We believe that the contribution is significantly higher than the possible bias in this situation. To address concerns we have mentioned this observation in Limitations subsection.
Comments 4: Table 2, the study of Uderzo should be omitted unless separate data are obtained for patients <18 years.
Table 4, the study of Treister should be omitted unless separate data are obtained for patients <18 years.
Response 4: Addressed in response 3.
Reviewer 3 Report
Comments and Suggestions for Authors
Although this study offers insightful information about dietary and supplement-based treatments for oral mucositis (OM) in pediatric oncology, a number of flaws restrict the study's power and applicability. The inclusion of solely randomized controlled trials may have eliminated pertinent observational or pilot studies that could have offered helpful context, especially given the paucity of pediatric data, even if the review complied with PRISMA criteria. It is more difficult to make meaningful syntheses and comparisons because to the variety in intervention kinds, doses, administration routes, and outcome measures among trials, which makes it harder to draw definitive findings. Small sample numbers and brief follow-up periods were probably included in several of the included trials, which limited statistical power and the evaluation of long-term safety or efficacy.Additionally, it doesn't seem like the review performs a quantitative meta-analysis or subgroup analysis, which would have improved the synthesis of the data and revealed possible moderators of treatment impact. Concerns regarding reproducibility and the inclusion of studies of lesser quality are also brought up by the dependence on resources like Google Scholar. Furthermore, the study skimps on quality evaluation techniques, publication bias, and risk of bias—all of which are crucial for assessing the trustworthiness of the evidence. Lastly, clinical applicability and safety interpretation are limited by the absence of information regarding possible negative effects or combinations of nutritional supplements.
The authors should consider to add some papers Aquino et al., 2005 , Widjaja et al., 2020 and Megari K., Katsarou D.V., Mantsos, E. Miliadi, V., Kosmidou, E., Argyriadi, A., Papadopoulou, S., Argyriadis, A., Toki, E.I., & Sofologi, M., (2025). “ Quality of Life and Anxiety of Adolescents With Cancer and Their Parents: Neurodevelopmental Implications in Adolescence.” International Journal of Developmental Neuroscience 85, no. 5: e70040. https://doi.org/10.1002/jdn.70040.
Author Response
Thank you for reviewing our manuscript. Bellow you will find our response and we hope that you will find it satisfactory.
Comments 1: Although this study offers insightful information about dietary and supplement-based treatments for oral mucositis (OM) in pediatric oncology, a number of flaws restrict the study's power and applicability. The inclusion of solely randomized controlled trials may have eliminated pertinent observational or pilot studies that could have offered helpful context, especially given the paucity of pediatric data, even if the review complied with PRISMA criteria. It is more difficult to make meaningful syntheses and comparisons because to the variety in intervention kinds, doses, administration routes, and outcome measures among trials, which makes it harder to draw definitive findings. Small sample numbers and brief follow-up periods were probably included in several of the included trials, which limited statistical power and the evaluation of long-term safety or efficacy
Response 1: The above concerns were addressed in the Limitations subsection.
Comments 2: Additionally, it doesn't seem like the review performs a quantitative meta-analysis or subgroup analysis, which would have improved the synthesis of the data and revealed possible moderators of treatment impact.
Response 2: This is possibly a misinterpretation, as subgroup analysis has been performed and can be easily differentiated by Results Subsection Titles such as: Systemically Administered Nutrient Supplementation regarding 7 out of the 20 studies; Topically Applied Nutritional Agents regarding 9 of the included studies.
A systematic review can be called a meta-analysis only when it includes a quantitative statistical synthesis as you have pointed out.
On the other hand the purpose of a Systematic Review is that of comprehensive and systematic summarization of relevant evidence on a specific objective, realising a qualitative synthesis.
While it draws more powerful conclusions, a Meta-Analysis needs to statistically combine numerical data from comparable studies to produce secondary data analysis and pooled effect estimates. Meta-Analyses require sufficiently homogenous outcome data can be statistically aggregated. This is just not the case with the objective and context this systematic review since the various agents and various measurements in effect of said agents. The need for better quality studies to in turn fuel more robust secondary data analysis such as meta-analysis has been stated in Future Research subsection. (R592-R594)
Comments 3: Concerns regarding reproducibility and the inclusion of studies of lesser quality are also brought up by the dependence on resources like Google Scholar.
Response 3: The Google Scholar source represents the smallest input (1.86%). Concerns of similar nature have been raised by Reviewer 2, and for transparency we are able to provide the search query used for each engine.
Comments 4: Furthermore, the study skimps on quality evaluation techniques, publication bias, and risk of bias—all of which are crucial for assessing the trustworthiness of the evidence.
Response 4: With al due respect, we believe this is a serious oversight, as study inclusion process has been described in the methods section, publication bias was mentioned in the limitations section. Regarding quality as well as risk of bias assessment, were already performed and were present in the supplementary material. This mention was present since the first version of the manuscript in subsection 2.6 of the Methods Section. Now the supplementary material contains the latest version of risk of bias assessment in colour-blind friendly palette.
Comments 5: Lastly, clinical applicability and safety interpretation are limited by the absence of information regarding possible negative effects or combinations of nutritional supplements.
Response 5: Combination of nutritional supplements in studies which evaluate efficacy may render the exact origin of the effect uncertain. Even so some of the studies look into various options which may be consisting of more than 1 suplement such as Traumeel S oral rinse (homeopathic complex of 14 herbs/minerals) used by Oberbaum et al. or Oberbaum et al. which evaluates both the efficacy of one nutritional agen (Honey) as well as a combination (honey, olive oil, propolis and beeswax). This information was already available in Table 1 of the manuscript.
Comments 6: The authors should consider to add some papers Aquino et al., 2005 , Widjaja et al., 2020 and Megari K., Katsarou D.V., Mantsos, E. Miliadi, V., Kosmidou, E., Argyriadi, A., Papadopoulou, S., Argyriadis, A., Toki, E.I., & Sofologi, M., (2025). “ Quality of Life and Anxiety of Adolescents With Cancer and Their Parents: Neurodevelopmental Implications in Adolescence.” International Journal of Developmental Neuroscience 85, no. 5: e70040. https://doi.org/10.1002/jdn.70040.
Response 6: Aquino et al., 2005 - already present as one of the included studies (second study in chronological order) in our systematic review.
Widjaja et al., 2020 – We apologise for the mistake. This study was returned by the search engine (Scopus) but was incorrectly excluded as it does not explicitly state in the abstract the RCT methodology, which is only implied. This was an oversight. The study is now included and all sections of the manuscript received changes.
Regarding the last suggested study we consider it to be partially correlated with matters discussed in our paper and was cited in the General Considerations subsection of the Discussion.
Round 2
Reviewer 2 Report
Comments and Suggestions for Authors
The paper has improved, although the problems with the inclusion criteria remain. Now it is written: Children and adolescents (mainly ≤18 years, excepting studies that covered paediatric patients and young adult patients who initiated treatment in childhood and were included in paediatric cohorts up to 25 years of age). That means that not all patients have to be ≤18 years thus also studies with older patient can be included as well as patients of 18 years that will be continued to treated in the period after their 18 year of birth, thus in fact when they are adolescent. So the study will be mixed up by patients who fullfill the peadiatric criteria (the 6 points mentioned why children differ from adults) and patients who are in fact already adults. This would not be a problem when the studies of Uderzo and Treister would be omitted. The other studies indeed only study children, while Uderzo and Treister also included (near) adults. This omission was also not mentioned in the limitations of this study.
Furthermore, the technical issues with the search terms should be mentioned, this could be an explanation why the difference in the number of papers could so greatly differ within the search engines.
Author Response
Reviewer 2
Comments 1: The paper has improved, although the problems with the inclusion criteria remain. Now it is written: Children and adolescents (mainly ≤18 years, excepting studies that covered paediatric patients and young adult patients who initiated treatment in childhood and were included in paediatric cohorts up to 25 years of age). That means that not all patients have to be ≤18 years thus also studies with older patient can be included as well as patients of 18 years that will be continued to treated in the period after their 18 year of birth, thus in fact when they are adolescent. So the study will be mixed up by patients who fullfill the peadiatric criteria (the 6 points mentioned why children differ from adults) and patients who are in fact already adults. This would not be a problem when the studies of Uderzo and Treister would be omitted. The other studies indeed only study children, while Uderzo and Treister also included (near) adults. This omission was also not mentioned in the limitations of this study.
Response 1: Limitations section previous statement: “The presence of young adults in largely paediatric populations in a small portion of the included studies is also worth keeping in mind.” has now been extended to better explain the limitation as follows: “Also, a degree of age heterogeneity, concerns the inclusion of two studies (Treister et al. and Uderzo et al.) whose cohorts extended slightly beyond the strict paediatric age thresh-old (13.7 (4.0–21.9) years and 8.1 (0.4-18.6) respectively) [43,49].”
Comments 2: Furthermore, the technical issues with the search terms should be mentioned, this could be an explanation why the difference in the number of papers could so greatly differ within the search engines.
Response 2: Differences in the number of records retrieved across databases reflect technical and structural variations in search engine functionality itself rather than inconsistencies in the strategy or issues with the search terms. Scopus indexes a broader range of journals and conference papers and allows TITLE-ABS-KEY searches across all indexed metadata, which explains its higher yield. Google Scholar provides limited control over field-specific queries. This may be useful in scientific oriented searches but not narrow, targeted queries such as the one we used. In the example of our study our study, using the same search query as for Scopus results in fewer academic results, as a consequence of database-specific characteristics and search engine limitations, rather than an issue with the search terms.
Reviewer 3 Report
Comments and Suggestions for Authors
Nothing further
Author Response
We appreciate you previous suggestions and current support.
Kind regards.